# Phosphoproteome Reveals Extracellular Regulated Protein Kinase Phosphorylation Mediated by Mitogen-Activated Protein Kinase Kinase-Regulating Granulosa Cell Apoptosis in Broody Geese

**DOI:** 10.3390/ijms241512278

**Published:** 2023-07-31

**Authors:** Shuai Zhao, Tiantian Gu, Kaiqi Weng, Yu Zhang, Zhengfeng Cao, Yang Zhang, Wenming Zhao, Guohong Chen, Qi Xu

**Affiliations:** 1Jiangsu Key Laboratory for Animal Genetic, Breeding and Molecular Design, Yangzhou University, Yangzhou 225009, China; zhaoshuai73@163.com (S.Z.);; 2Joint International Research Laboratory of Agriculture and Agri-Product Safety, The Ministry of Education of China, Yangzhou University, Yangzhou 225009, China

**Keywords:** goose, phosphoproteomic, ERK, brooding

## Abstract

Geese have strong brooding abilities, which severely affect their egg-laying performance. Phosphorylation is widely involved in regulating reproductive activities, but its role in goose brooding behavior is unclear. In this study, we investigated differences in the phosphoprotein composition of ovarian tissue between laying and brooding geese. Brooding geese exhibited ovarian and follicular atrophy, as well as significant oxidative stress and granulosa cell apoptosis. We identified 578 highly phosphorylated proteins and 281 lowly phosphorylated proteins, and a KEGG pathway analysis showed that these differentially phosphorylated proteins were mainly involved in cell apoptosis, adhesion junctions, and other signaling pathways related to goose brooding behavior. The extracellular regulated protein kinase (ERK)–B-Cell Lymphoma 2(BCL_2_) signaling pathway was identified as playing an important role in regulating cell apoptosis. The phosphorylation levels of ERK proteins were significantly lower in brooding geese than in laying geese, and the expression of mitogen-activated protein kinase kinase (MEK) was downregulated. Overexpression of MEK led to a significant increase in ERK phosphorylation and *BCL2* transcription in H_2_O_2_-induced granulosa cells (*p <* 0.05), partially rescuing cell death. Conversely, granulosa cells receiving MEK siRNA exhibited the opposite trend. In conclusion, geese experience significant oxidative stress and granulosa cell apoptosis during brooding, with downregulated MEK expression, decreased phosphorylation of ERK protein, and inhibited expression of *BCL2*.

## 1. Introduction

Broodiness is a maternal behavior that alternates with laying eggs during a hen’s laying cycle. Although broodiness is essential for ensuring natural reproduction, the widespread use of artificial incubation techniques has revealed its negative impact on egg production. When hens become broody, they stop laying eggs and focus on hatching their eggs, which hinders the goal of egg production [1]. Geese are known to be highly broody, with over 90% of goose species exhibiting broodiness, such as Zhedong white geese and Magang geese [2]. The strong broodiness of geese poses a significant challenge to the development of the goose industry in China. Therefore, a comprehensive study of the physiological mechanisms underlying broodiness is necessary to enhance the egg production performance of geese.

The ovary is an important organ in the female reproductive system, and its basic role is to produce and release eggs, as well as to secrete sex hormones to regulate the reproductive cycle [3]. However, during the brooding cycle of geese, there is an increase in reactive oxygen species (ROS) in the ovary [4], which may lead to damage and apoptosis of follicles and granulosa cells [5]; granulosa cells being the main cell type of follicles, apoptosis of granulosa cells and the occurrence of ovarian dysfunction may lead to egg production arrest in geese [6]. Therefore, the effective control of the oxidative stress and apoptosis of granulosa cells is essential to slow down the brooding period and increase egg production.

ERK is involved in both intrinsic and extrinsic apoptotic pathways [7,8,9]. In the intrinsic pathway, ERK can activate members of the pro-apoptotic Bcl-2 family [10], leading to the release of cytochrome c and the activation of caspases [11], and in the extrinsic pathway, ERK can regulate the activation of the death receptor Fas [12]. ERK is affected by various epigenetic modifications, including DNA methylation and histone modifications [13,14], and these modifications can affect its expression. For example, phosphorylation of Erkl/2 has a protective effect on the survival of rat axons and neurons [15,16], and conversely, phosphorylation of ERK and FAK/AKT can induce apoptosis [17]. Thus, epigenetic modifications of ERK can have both pro- and anti-apoptotic effects depending on the specific situation [18].

As an epigenetic modification, protein phosphorylation modification can affect many cellular processes, including cell metabolism, signal transduction, and apoptosis [19,20,21]. As two different physiological behaviors in the egg-laying cycle of geese, brooding and egg-laying are typical epigenetic phenomena that should be regulated by epigenetic modifications, but the specific regulatory mechanisms are still unclear. This study will utilize LC-MS/MS-based proteomic analysis to investigate changes in protein phosphorylation in the ovaries of geese during the laying and brooding periods and provide data to further explore the molecular mechanisms of goose brooding behavior in an attempt to solve the problem of low production performance.

## 2. Results

### 2.1. Morphological Detection and Antioxidant Activity of Ovaries in Broody Geese

The ovarian morphology of the laying geese and broody geese was observed. The results revealed that the sizes, lengths, and widths of the ovaries of the broody geese were significantly lower than those of laying geese (*p* < 0.05). Additionally, the cortex and medulla of the ovaries of the broody geese were shrunken, follicles were severely degenerated, and pre-ovulatory follicles were absent (Figure 1A). The average volume of the ovaries of the broody geese was approximately 8 cubic centimeters, while the corresponding value was around 160 cubic centimeters in the laying geese (Figure 1B). Also, we used transmission electron microscopy (TEM) and counted the apoptotic cells in electron microscopy images of the same area. The results showed that the ovaries of the broody geese had significantly more apoptotic cells than those of the laying geeseovaries (Figure 1C,D). Furthermore, we measured the content of intracellular reactive oxygen species (MDA, GSH, H_2_O_2_) and the activity of antioxidant enzymes (SOD, CAT, GSH-PX) in the ovaries between the laying and broody geese. The results demonstrated that the levels of intracellular reactive oxygen species and the activity of antioxidant enzymes were significantly increased in broody geese compared with the laying geese (Figure 1E).

### 2.2. MS-Based Quantitative Proteomic and Phosphoproteomic Characterizations of the Ovaries of Laying and Broody Geese

In this study, we aimed to quantify the protein and modification site levels in repeated samples and determine the fold change (FC) differences between the two groups. We set thresholds for upregulation at a fold change greater than 1.2 and for downregulation at less than 1/1.2 (*p* < 0.05). Both our phosphoproteomics and proteomics principal component analyses (PCAs) showed significant differences in the major protein components of the ovaries between the laying and brooding periods, indicating that significant changes in ovarian protein and phosphorylation levels occur during the brooding period of the goose (Figure 2I). Of the 4161 quantifiable proteins identified, 281 (7%) were upregulated and 422 (10%) were downregulated (Figure 2B,C). Of the 5707 quantifiable phosphorylation sites and 1585 phosphorylation-modified proteins, 19% of the upregulated phosphorylation sites and 9% of the downregulated sites mapped to 36% and 18% of the proteins, respectively (Figure 2D,E). The specifics of the differentially expressed proteins and phosphorylation sites are shown on the heat map (Figure 2F). Most of the phosphorylation modifications (86%) occurred on serine, 13% on threonine, and only 1% on tyrosine (Figure 2G). Integrative analysis of phosphoproteomics and proteomics data showed that more than 36% of the quantifiable phosphorylation-modified proteins (8.93% with both phosphorylation and protein level regulation and 27.53% with only phosphorylation modification regulation) exhibited significant changes in phosphorylation modification levels (Figure 2H). Of the proteins that were simultaneously regulated at the protein and phosphorylation levels, 26.4% exhibited increased (16%) or decreased (10.4%) levels (Figure 2J). Proteins that were only regulated by phosphorylation modification will be further analyzed to exclude interference caused by protein changes. Overall, our results showed that 17% of the measured proteins and 54% of the phosphoproteins changed in the ovaries during the laying and breeding periods (Figure 2C,E), with a greater proportion of changes observed in phosphorylated proteins. Our results emphasize the importance of joint analysis of protein and phosphoprotein changes to fully understand the molecular changes in the ovarian cycle of geese. 

### 2.3. Identification of Signaling Pathways and Key Substrate of Differential Abundance Phosphoproteins between Laying and Broody Geese

To uncover potential conserved phosphorylation protein motifs in goose ovaries, phosphorylation sites were identified, and a motif analysis was performed. A total of 139 putative phosphorylation patterns were found, including 123 serine patterns and 16 threonine patterns. The amino acids surrounding the phosphorylation sites were mainly aspartic acid, glutamic acid, arginine, serine, lysine, and proline (Figure 3E). The most frequently occurring motifs, such as “xxxRSx_S_Pxxxxx” and “xxxxxx_S_EEExxx” and “xxxxxx_S_DxExEx” (Figure 3F), may play a role in goose brooding behavior.

We then investigated the biological processes that underwent the greatest changes from the laying period to the brooding period in goose ovaries, and at which molecular levels. Proteins involved in autophagy and apoptosis were enriched in the KEGG analysis of the goose ovary tissue proteome, which was mainly enriched in the focal adhesion, Phagosome, and Pi3K-Akt signaling pathways (Figure 3A). Furthermore, in terms of phosphorylation signaling, the focal adhesion and regulation of actin cytoskeleton pathways were significantly enriched in the brooding period, as revealed by the KEGG analysis (Figure 3B). Protein interaction analysis of proteins with significantly altered phosphorylation levels revealed that MAPK1 (ERK) may be a key protein during goose rearing (Figure 3C), with a more than 2-fold increase in the phosphorylation levels of the ERK protein (Figure 3D). It was also found that the phosphorylation changes in the ERK protein occurred simultaneously on the pathways of focal adhesion and regulation of the actin cytoskeleton. Therefore, we propose that the phosphorylation of ERK protein is closely related to the occurrence of goose maternal behavior.

### 2.4. Functional Validation of Specific Kinase (MEK)

In this study, we observed that the ERK protein of the ovaries was hypo-phosphorylated in the broody geese compared with the laying geese, but its mechanisms were unclear. We used iGPS software to predict the ERK protein kinase and identified MAP2K1 (MEK) as the kinase that catalyzes ERK phosphorylation at Threonine 188 (Figure 4A). Moreover, a significantly lower mRNA expression of *MEK* in the ovaries was detected in broody geese in comparison with the laying geese (Figure 4B). Also, we detected that the phosphorylation levels of ERK protein were significantly lower in broody geese than those in the laying geese (Figure 4C,D).

### 2.5. Regulation of Granulosa Cell Apoptosis by MEK-Mediated Phosphorylation of ERK

To further examine the effects of kinase MEK on ERK phosphorylation and the downstream gene *BCL2* expression in H_2_O_2_-induced granular cells (GCs), we found that GCs that were inducted with 100 μmol/L H_2_O_2_ and a large number of cells were apoptotic (Figure 5D). The overexpression of MEK resulted in a significant increase in ERK phosphorylation in H_2_O_2_-induced GCs (*p* < 0.05). Conversely, MEK siRNA led to a significant decrease in ERK phosphorylation (Figure 5A,B). These results indicate that the expression of MEK promoted ERK protein phosphorylation.

Next, we investigated the effects of the overexpression/knockdown of MEK on the downstream gene *BCL2* expression and cell survival in H_2_O_2_-induced GCs. We found the overexpression of MEK contributed to increased *BCL2* transcription (Figure 5C), which rescued cell death to some extent (with an increase in the average value, but without reaching a significant level). Instead, GCs receiving MEK siRNA showed the opposite trend (Figure 5C,D).

## 3. Discussion

Protein phosphorylation modifications may have a significant impact on cell apoptosis. However, the apoptosis of granulosa cells in goose ovaries can lead to ovarian degeneration, a decrease in egg production, and affect the development of the goose industry. In this study, we compared the size and morphology of the ovaries of Zhedong white geese during the laying and brooding periods. As the geese transitioned from the laying period to the brooding period, their ovaries significantly decreased in size, with follicles regressing and the disappearance of hierarchical follicles, which is consistent with the changes observed in the ovaries of other poultry during the brooding period [22,23]. When we observed the ultrastructure of the ovaries, we found that a large number of granulosa cells were apoptotic during the brooding period [24], and the nuclei showed wrinkling and deformation, but the cell membranes were intact, and some of them showed foaming or apoptotic vesicles [25], while the apoptotic cells in the ovaries during the laying period were significantly fewer than those in the brooding period, and the antioxidant enzyme activities (SOD, CAT, GSH-PX) and oxidized free radicals (MDA, GSH, H_2_O_2_) in the ovaries of geese during the brooding period were detected to be significantly higher. ROS can cause oxidative damage to cells, including the induction of apoptosis [26,27]. Therefore, we suggest that follicular atresia and apoptosis are closely related to oxidative stress in goose broods.

Since the strong brooding behavior of geese affects their egg-laying performance, this study was conducted to identify and screen ovarian proteins and phosphoproteins during the laying period and the brooding period using the LC-MS/MS technique on Zhedong white geese with strong brooding behavior. Our biological replicate Pearson correlation coefficients were all around 0.7, which is in a plausible range; also, our PCAs of the proteomics and phosphoproteomics data were highly aggregated, clearly classifying the laying period and brooding ovaries and representing good quantitative reproducibility from the side; likewise, in agreement with other MS-based studies [28], the majority of our phosphorylation events occurred on serine residues (86%), followed by threonine and tryptophan. A fraction of the quantifiable proteins were significantly modulated at the phosphorylation and protein levels, and it could not be determined that phosphoproteins acted exclusively; therefore, we analyzed an additional 27.53% of proteins with significant regulation of only the phosphorylation levels to exclude the effect of protein changes [29].

In our results, the regulated proteins were mainly enriched in the PI3K-Akt signaling pathway. The PI3K/Akt pathway is a key regulator of survival during cellular stress [30]. The PI3K/Akt pathway is thought to be associated with tumors because of the inherent stressful environments in which they exist [31,32]. The PI3K-Akt signaling pathway is essential for many physiological and pathological conditions, such as cell proliferation, differentiation, and survival [32], and in terms of cell survival, Akt/PKB can inactivate pro-apoptotic factors such as Bad and caspase-9 [33,34]. The PI3K/Akt/mTOR pathway is activated in approximately 70% of ovarian cancers [35]. The regulation of local adhesion and the actin cytoskeleton, which are significantly enriched in phosphorylated proteins, plays an important role in pathways related to cell motility, bone system development, cell cycle and cell proliferation, and cell survival [36,37]. The differentially expressed phosphoproteins in the focal adhesion signaling pathway are vinculin and ERK proteins. Vinculin is a cytoskeletal protein and adhesive patch component protein, mainly associated with cell adhesion, extension, and motility [38,39,40]; activation of ERK is widely associated with cell proliferation, differentiation, apoptosis, and transcriptional functions [41,42,43]. We also performed protein interaction analysis on proteins with large changes in phosphorylation levels and found that ERK was the key protein and ERK translational modifications can regulate apoptosis by regulating pro-apoptotic proteins; for example, phosphorylated ERK increases the expression of *BCL2* in cells [44]. MEK is a rare bispecific kinase that can deactivate ERK through the phosphorylation of two regulatory sites (Tyr 204/187 and Thr 202/185) [45]. MEK acts as a phospho-kinase for ERK proteins, and upregulation of MEK increases the phosphorylation level of ERK [46], which is the same as we predicted.

Our results show that ovarian ERK phosphoprotein is inhibited during the brooding period, together with an increased level of oxidation in the ovary, and some studies have shown that antioxidant treatment can increase ERK phosphorylation [47]; therefore, we speculate that the elevation of oxidative factors is the main reason for the inhibition of ERK phosphorylation. Although the precise mechanisms are not well defined, evidence has suggested that ROS generations are upstream events leading to ERK activation [48,49,50]. We used H_2_O_2_ to construct an oxidative stress model [51,52] and treat goose GCs with reduced ERK phosphorylation levels and *BCL2* expression. Similarly, MEK acts as a phosphorylated kinase of ERK, and MEK expression positively regulates ERK phosphorylation [53]. As our results show, ERK phosphorylation levels and *BCL2* expression are positively regulated by MEK. *BCL2* is an anti-apoptotic protein that enables cells to resist apoptosis [54,55]. It has been demonstrated that H_2_O_2_ can induce the elevation of endogenous pro-apoptosis-related molecules (Bax, Bak) and a decrease in anti-apoptotic molecules (Bcl-2, Bcl-xL) and regulate apoptosis in ovarian granulosa cells through the ROS-JNK-p53 pathway [56]. Therefore, we examined the activity of GCs after the knockdown and overexpression of MEK treatment and H_2_O_2_ treatment to show that the activity of H_2_O_2_-treated and knockdown MEK GCs was significantly reduced.

This study revealed the differences in oxidative stress levels in the ovaries of geese during the breeding and laying periods and explored the role of protein phosphorylation in the process of goose breeding. However, this study also has some technical limitations; changes in oxidative stress and GC apoptosis and protein phosphorylation can only be detected at the cellular level, rather than at the level of the entire organ in vivo, as the signal transduction structure is not intact after ovarian granulocyte separation. In addition, oxidative stress and nesting behavior in geese occur simultaneously, and we do not have a method to establish their causal relationship, although this study found that changes in the oxidative environment can alter the phosphorylation levels of ERK protein. Therefore, we are attempting to alter geese’s nesting behavior by feeding them antioxidant-rich foods.

## 4. Materials and Methods

### 4.1. Animal and Sample Collection

Zhedong white geese have strong brooding behavior; thus, 350-day-old Zhedong white geese were chosen as the experimental subjects for this study. The laying geese were identified by the artificial holding of eggs on their bellies, while the brooding geese were confirmed through continuous manual observation. If the geese stayed in their nests for more than 2 consecutive days, we classified them as broody. The geese were from the breeding farm of Jiangsu Lihua Animal Husbandry Co., Ltd. in China. The three laying geese and three brooding geese were selected and anesthetized using sodium pentobarbital. The ovarian tissues were rapidly collected; some were put in freezing tubes and frozen in liquid nitrogen for phosphoproteome and quantitative real-time PCR (qPCR), and some were put in electron microscope fixative for transmission electron microscopy.

### 4.2. Morphological Detection and Antioxidant Activity of Ovaries

The water displacement method was used to measure the volume of each ovary. A 500 mL graduated cylinder was prepared, and 200 mL of PBS was added to it. The intact ovary was taken from the goose and placed in the graduated cylinder, ensuring that the liquid completely covered the ovary. The initial and final readings were recorded, and the difference in volume was calculated as the volume of the ovary. Next, a 1 mm^3^ piece of ovarian follicle tissue was quickly taken and placed in electron microscope fixative to prepare ultrathin sections for the transmission electron microscope. The representative region of the ultrathin section was observed using a Hitachi 7800 electron microscope (Japan) for image analysis. At total of 100 mg of tissue was accurately weighed and placed in a homogenization tube, and 900 mL of physiological saline was added. The tissue was homogenized under ice-cold conditions to prepare a 10% tissue homogenate, which was then centrifuged at 2500 rpm for 10 min to obtain the supernatant. The supernatant was diluted to the optimal sampling concentration with physiological saline, and then the MDA, GSH, H_2_O_2_, SOD, CAT, and GSH-PX assay kits provided by Nanjing Jiancheng were used to detect oxidative factors in the goose ovary.

### 4.3. Proteome and Phosphoproteome Analyses

#### 4.3.1. Protein Extraction

After removing the ovarian tissue samples from the ultra-low-temperature freezer at −80 °C, an appropriate amount of each sample was weighed and placed into a homogenization tube, followed by the addition of RIPA protein lysis buffer, protease inhibitor, and phosphatase inhibitor (in a ratio of 98:1:1), and subjected to ultrasonic lysis under light-avoiding conditions. The mixture was then centrifuged (at a temperature of 4 °C, a centrifugation speed of 12,000× *g*, and a centrifugation time of 10 min) to remove cell debris. Finally, the supernatant was transferred to a new centrifuge tube, and the protein concentration was measured using a BCA protein assay kit.

#### 4.3.2. Protein Digestion and TMT Labeling

Take equal amounts of protein samples and digest them enzymatically. Adjust the sample volume to the same level using a lysis buffer. Add 20% TCA to precipitate the proteins and wash the precipitate three times with acetone. Then add TEAB with a final concentration of 200 mM. Add trypsin at a ratio of 1:50 (trypsin:protein) and digest overnight. Add DTT with a final concentration of 5 mM for reduction, and react at 56 °C for 30 min. Then add IAA with a final concentration of 11 mM and incubate in the dark at room temperature for 15 min. Desalt the trypsin-digested peptides using Strata X C18 (Phenomenex), and then perform vacuum freeze-drying. Dissolve the peptides in 0.5 M TEAB and label them according to the instructions of the TMT reagent kit.

#### 4.3.3. LC-MS/MS Analysis and Database Search

In this experiment, the peptides were separated using the EASY-nLC 1200 UPLC system, ionized in the NSI source (2.1 kV), and analyzed using the Q Exactive™ HF-X mass spectrometer. The first-level mass spectrum scanning range was set to 350–1600 m/z with a resolution of 60,000, while the second-level mass spectrum scanning range was set to a fixed starting point of 100 *m*/*z* with a resolution of 30,000. Maxquant (v1.6.15.0) software was used to search the second-level mass spectrum data. The search parameters were set as follows: the database was Anse cygnoides 8845 NCBI PRJNA183603 20201209.fasta (31,811 sequences), and the reverse library was added to calculate the false discovery rate (FDR) caused by random matches.

#### 4.3.4. Bioinformatics Analysis

In this study, the identified proteins were annotated and analyzed for enrichment using eggnog-mapper software (v2.0) and the eggNOG database for gene ontology (GO) annotation and enrichment analysis, the PfamScan tool and the InterPro database for protein domain annotation and enrichment analysis, and the Kyoto Encyclopedia of Genes and Genomes’ (KEGG) pathway database for protein pathway annotation and enrichment analysis. A two-tailed Fisher’s exact test was employed for each category to test the enrichment of the differentially expressed proteins against all identified proteins, and a *p*-value of less than 0.05 was considered significant. Protein–protein interaction and motif analyses were then conducted using the Jingjie Bioinformatics Cloud Platform (http://www.ptmbiolab.com accessed on 22 December 2020) to identify key regulatory modified proteins under specific experimental conditions.

### 4.4. Isolation and Identification of Goose Ovarian Granulosa Cells

Goose follicles were obtained and processed by removing the outer follicle membrane’s connective tissue, puncturing the follicles to release the follicular fluid, and then washing and cutting them up. The follicle tissue was then digested by adding 20 ng/mL of collagenase type II and incubating for 15 to 20 min at 38.5 °C. Tissue debris that were not fully digested were removed by filtration through a 200-mesh screen. The resulting filtrate was transferred to a new centrifuge tube and centrifuged at 800× *g* for 10 min. After removal of the supernatant, the granulosa cells were resuspended in serum containing DMEM and incubated in a cell culture chamber at 38.5 °C with 5% CO_2_. Well-growing goose ovarian follicle granulosa cells were evenly seeded in a 12-well plate, and when the cell fusion degree reached 60–70%, indirect immunofluorescence staining identification (anti-FSHR) was performed. A total of 100 μmol/L H_2_O_2_ was added to establish an in vitro oxidative stress model of goose ovarian follicle granulosa cells [6].

### 4.5. Obtaining Overexpression Vectors and MEK siRNA, and Transfecting Them

Using cDNA from the spleen of the Zhedong white geese as a template, the full length of the MEK was amplified (F: ccaagctggctagttaagcttATGCCGGCCAAGAGGAAG, R: ccacactggactagtggatccCATGGCAGCGCGGGTGGG), double-digested, and inserted into the pcDNA3.1-Flag expression vector to construct the pcDNA3.1-Flag-MEK expression plasmid (MEK-flag). The connecting product was transformed into E. coli DH5a, and positive recombinant plasmids were selected for sequencing and subsequent experiments after successful alignment. MEK siRNA (si-MEK, F: GCAAGAUUUCCAGGAGUUUTT, R: AAACUCCUGGAAAUCUUGCTT), RNAi positive control, negative control, and FAM-labeled negative control were all purchased from Shanghai GenePharma Co., Ltd. (Shanghai, China). According to the manufacturer’s instructions, transfection was performed when the cell density reached 70%. LipofectamineTM 2000 was used for transient transfection of goose GCs, and transfection reagent mixture was added to continue the cell culture. After 6 h, the culture medium was replaced for subsequent experiments.

### 4.6. RNA Extraction, cDNA Synthesis, and Quantitative Real-Time PCR

TRIzol (Vazyme) was used to extract mRNA from goose tissues or cells. cDNA synthesis was performed using the FastKing gDNA Dispelling RT SuperMix kit, real-time fluorescence quantitative PCR was performed using the Cham^Q^TM SYBR^®®^ qPCR Master Mix, the QuantStudio Real-Time Fluorescence PCR System was used for quantitative analysis, and all primers were synthesized by Tsingke Biotechnology Co. (Beijing, China). The following primers were used: β-actin (F, 5′-GGAAATCGTGCGTGACATTA-3′; R, 5′-AGGAAGGAAGGCTGGAAGAG-3′), MEK (F: 5′-AGCCTTCCTCACGCAGA-3′, R: 5′-GCTCCATAGAAACCCACG-3′), and *BCL2* (F: 5′-GCCTGGATGACCGAGTA-3′, R: 5′-GCCATACAACTCCACGAA-3′). Model data were normalized to β-actin.

### 4.7. Capillary Western Blotting

The prepared samples, ladder, antibody dilutions, luminescent solution, diluted primary antibodies, and corresponding secondary antibodies were added to the designated wells of the assay plates, centrifuged, and put into the Protein Simple WES instrument for detection and analysis using Compass for SW software (Version 6.1.0). Antibodies used were Anti-ERK1 (phosphor T202) (Abcam#ab47310), Anti-ERK (Abmart# T40071), and β-actin (novus#NB600-532).

### 4.8. Statistical Analysis

All experiments in this study were performed in triplicate. Data were analyzed using paired comparisons with SPSS 25.0 (Chicago, IL, USA), including statistical data, qPCR results, and capillary Western blotting results, and presented as the mean ± standard error of the mean (SEM). Fisher’s exact test was used for two-tailed testing of protein enrichment levels for all identified proteins to determine differences in protein expression. Significance was determined when the *p*-values were less than 0.05 or 0.01 for each category.

## 5. Conclusions

In summary, compared to the ovaries of geese during the laying period, the oxidative stress levels of the ovaries of geese during the breeding period are significantly increased, and the protein phosphorylation is significantly changed, with the phosphorylation of ERK protein regulating granulocyte apoptosis through the MEK/ERK/BCL2 signaling pathway and participating in the regulation of goose breeding. Overall, this study revealed, to some extent, the molecular mechanism underlying goose breeding behavior at the protein level, provided insights into the mechanism of granulocyte apoptosis in geese, and emphasized the importance of oxidative stress and protein phosphorylation in this process. These findings may contribute to the development of strategies to improve goose egg production performance and promote the development of the goose industry.

## Figures and Tables

**Figure 1 ijms-24-12278-f001:**
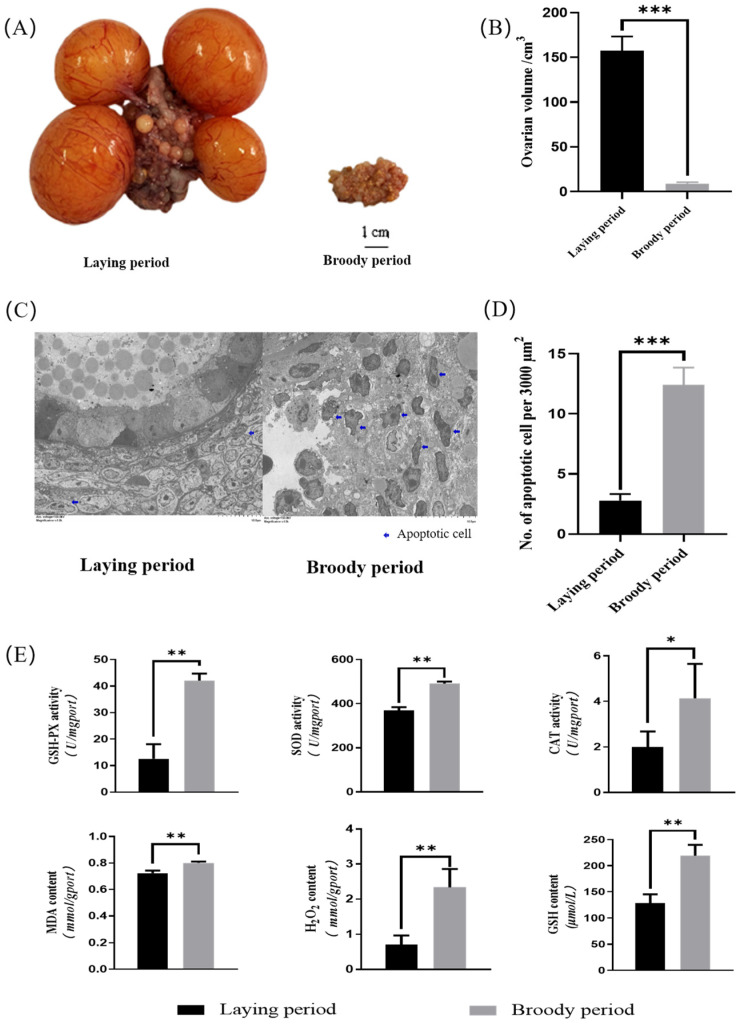
Morphological detection and antioxidant activity of ovaries in laying geese and broody geese. (**A**,**B**) Comparisons of the ovarian phenotype between the laying and brooding periods of Zhedong white geese. (**C**) Comparison of TEM images of Zhedong white geese ovaries during the laying and brooding periods. (**D**) Comparison of the number of apoptotic cells. (**E**) Analysis of the contents of tissue oxidative factors (MDA, GSH, H_2_O_2_) and the activity of antioxidant enzymes (SOD, CAT, GSH-PX) in the ovaries of Zhedong white geese during the laying and brooding periods. The error bars represent the standard error (SE) of three replicates. Significant differences were compared using Student’s *t*-test (n = 3; * *p* < 0.05, ** *p* < 0.01, *** *p* < 0.001).

**Figure 2 ijms-24-12278-f002:**
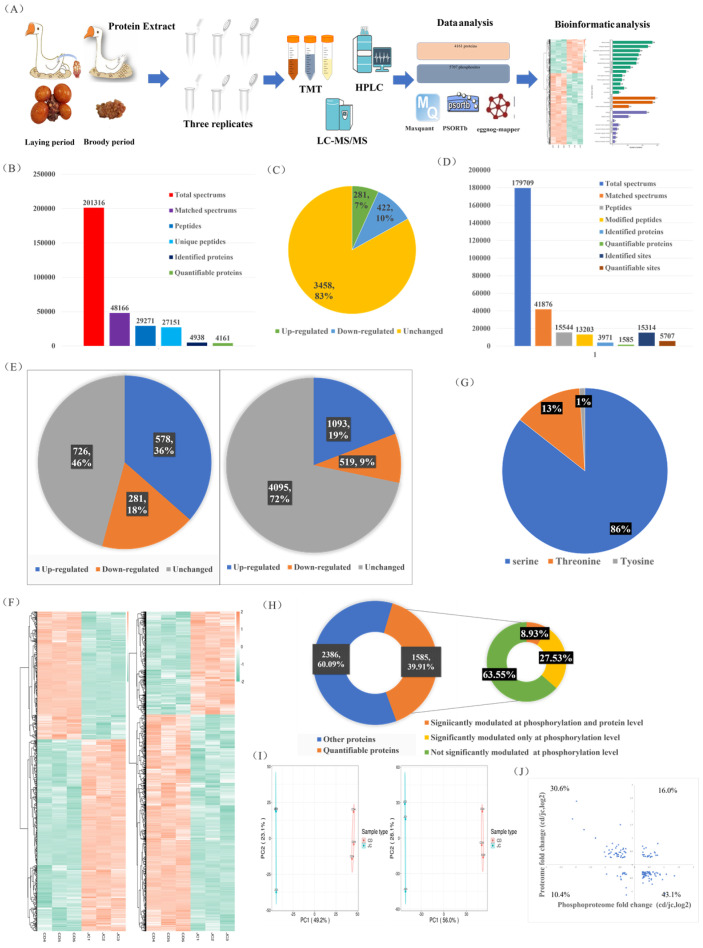
Integrated analysis of quantitative proteomics and phosphoproteomics in the ovaries of laying and broody geese. (**A**) Experimental workflow for analyzing the proteomic and phosphoproteomic profiles of the ovaries of geese during the laying and brooding periods. (**B**,**D**) Results of protein and phosphoprotein database searches. (**C**) Differential protein expression statistics. (**E**) Differential phosphorylation site and protein statistics. (**F**) Heat map of differentially expressed proteins and phosphorylated proteins. (**G**) Distribution of phosphorylation sites on serine, threonine, and tyrosine. (**H**) Proportion of quantifiable phosphorylated proteins (left) and correlation analysis of protein changes and phosphorylation level changes (right). (**I**) PCAs of proteomic (left) and phosphoproteomic (right) data. (**J**) Distribution of proteins with simultaneous changes in protein levels and phosphorylation levels.

**Figure 3 ijms-24-12278-f003:**
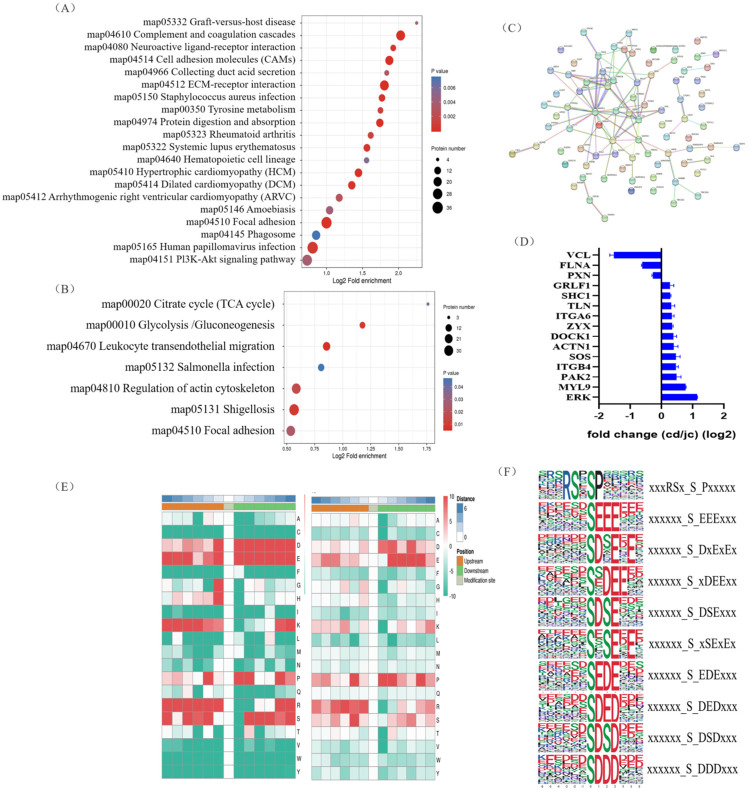
Identification of key protein (ERK) involved in goose nesting behavior through KEGG pathway enrichment analysis and protein–protein interaction network screening. (**A**,**B**) Bubble charts showing KEGG pathway enrichment of differentially expressed proteins (**A**) and differentially expressed phosphorylated proteins (**B**). (**C**) Protein–protein interaction network of differentially expressed phosphorylated proteins. (**D**) Partial fold changes of some differentially phosphorylated proteins. (**E**) Motif analysis of phosphorylation modification sites (serine, left; threonine, right). (**F**) The most frequently occurring motif near the phosphorylation modification sites based on motif analysis.

**Figure 4 ijms-24-12278-f004:**
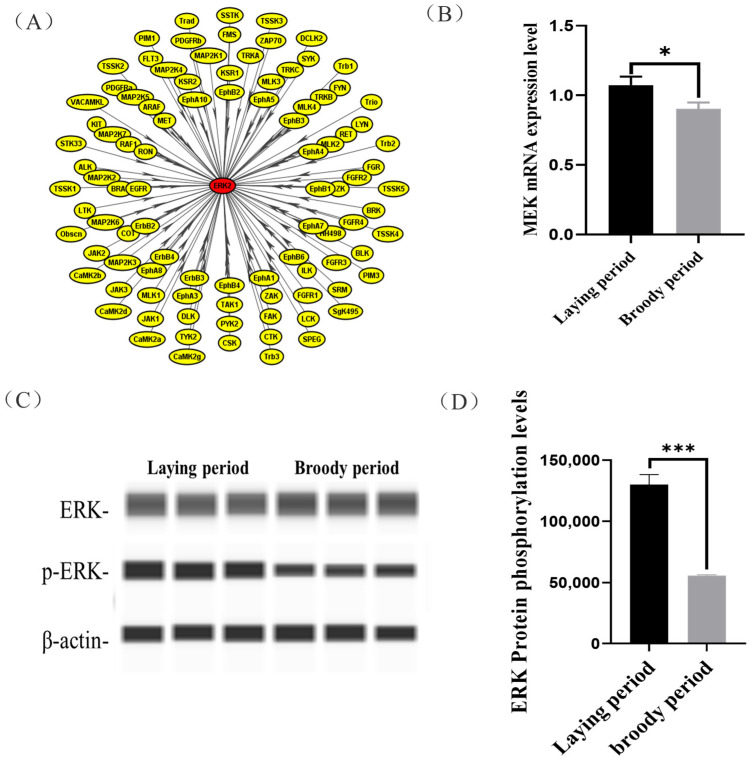
Prediction of the phosphorylation kinase of ERK(MEK) and its function validation. (**A**) We used iGPS software to predict ERK protein kinase. (**B**) The expression levels of MEK in the ovaries of laying and broody geese. (**C**,**D**) The expression levels of ERK protein and its phosphorylation status in the ovaries of laying and broody geese. The error bars represent the standard error (SE) of three replicates. Significant differences were compared using Student’s *t*-test (n = 3; * *p* < 0.05, *** *p* < 0.001).

**Figure 5 ijms-24-12278-f005:**
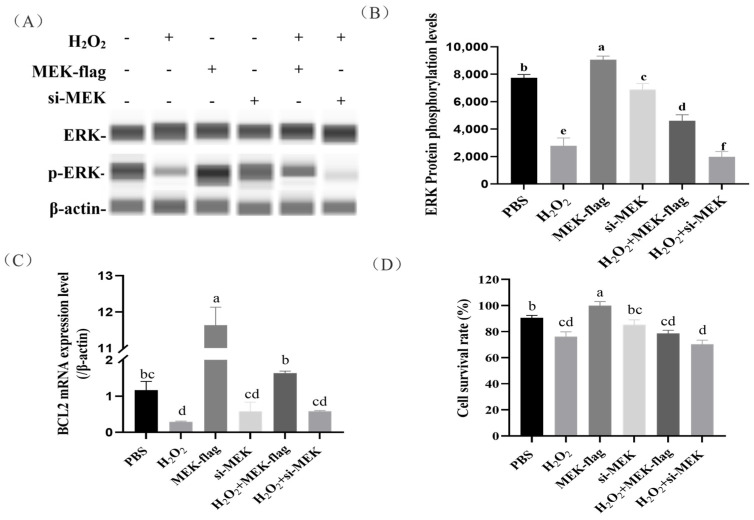
The effects of the overexpression/knockdown of MEK on downstream gene *BCL2* expression and cell survival in H_2_O_2_-induced GCs. (**A**,**B**) The effects of MEK on ERK protein phosphorylation were evaluated in H_2_O_2_-induced GCs. (**C**) The effects of the overexpression/knockdown of MEK on downstream gene *BCL2* expression in H_2_O_2_-induced GCs. (**D**) The effects of the overexpression/knockdown of MEK on cell survival rate in H_2_O_2_-induced GCs. The error bars represent the standard error (SE) of three replicates. Significant differences were compared using Student’s *t*-test, Groups sharing the same letter label are considered not significantly different (*p* < 0.05), whereas groups with different letter labels are regarded as exhibiting statistically significant differences.

## Data Availability

The data presented in this study are available in article.

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
