# Peer review of "Phosphoproteome Reveals Extracellular Regulated Protein Kinase Phosphorylation Mediated by Mitogen-Activated Protein Kinase Kinase-Regulating Granulosa Cell Apoptosis in Broody Geese"

_ijms, 2023, doi:10.3390/ijms241512278_

Round 1
Reviewer 1 Report
The research titled 'Phosphoproteome reveals ERK phosphorylation mediated by MEK regulating granulosa cells apoptosis in broody geese' addresses an important and timely topic. I found the article's subject matter to be fascinating, and I read the manuscript with great interest. The paper aligns well with the scope of the journal. However, I believe that in its current form, it has several shortcomings:
- Providing more comprehensive details about the study population: Specifically, it would be beneficial to include additional information on the geese population, such as the number of animals involved in the research, their age, body weight, and other relevant characteristics. Additionally, please include a brief description of the breed of geese selected and the rationale behind choosing this particular breed for the study.
- Ethical statement: I recommend including the Ethical statement in the M&M section of the manuscript to ensure transparency and compliance with ethical guidelines.
- Formatting: The subheading in Section 4.7 should not be in italics. Please ensure that the formatting is consistent throughout the manuscript.
- I suggest expanding the Discussion section to include study limitations and the practical implications.
Author Response
Response to Reviewer 1 Comments
Dear Reviewer,
We sincerely appreciate your review of our manuscript and valuable feedback. We have carefully considered your comments and made the necessary revisions and additions. Please find our point-by-point response below:
Point 1: Providing more comprehensive details about the study population:Specifically, it would be beneficial to include additional information on the geese population, such as the number of animals involved in the research,their age, body weight, and other relevant characteristics. Additionally, please include a brief description of the breed of geese selected and the rationale behind choosing this particular breed for the study.
Response 1: Thank you for your valuable suggestion. As per your advice, we have now included more comprehensive information about the study population in the "Materials and Methods" section. Specifically, we have provided details on the number of animals involved in the research and their age, among other relevant characteristics. Additionally, we have briefly explained the rationale behind selecting the specific breed of geese for our study. This breed was chosen due to its pronounced brooding behavior and significant morphological changes in the ovaries during the egg-laying and brooding periods. Consequently, each experimental group consisted of three 350-day-old Zedong White Geese as the subjects of our investigation (Lines 310-315).
Point 2: Ethical statement: I recommend including the Ethical statement in the M&M section of the manuscript to ensure transparency and compliance with ethical guidelines.
Response 2: We are truly grateful for your guidance on this matter, due to journal requirements, the ethical declaration is placed at the end of the manuscriptethical declaration is now included in the manuscript (Line 455-461).
Point 3: Formatting: The subheading in Section 4.7 should not be in italics. Please ensure that the formatting is consistent throughout the manuscript.
Response 3: We sincerely apologize for the oversight regarding the formatting issue. We have rectified this mistake in the manuscript, ensuring Section 4.7 is no longer in italics. Moreover, we have meticulously reviewed the entire manuscript to ensure consistency in formatting throughout (Lines 425-430).
Point 4: I suggest expanding the Discussion section to include study limitations and the practical implications.
Response 4: In the discussion section, we explore the role of protein phosphorylation on the ovary during goose reproduction and also illustrate the technical limitations of the study, the changes in oxidative stress and GCs apoptosis and protein phosphorylation can only be detected at the cellular level, rather than at the level of the entire organ in vivo, as the signal transduction structure is not intact after ovarian granulocyte sepa-ration. In addition, oxidative stress and nesting behavior in geese occur simultaneously, and we do not have a method to establish their causal relationship, although this study found that changes in the oxidative environment can alter the phosphorylation level of ERK protein. (Line298-307)
We sincerely appreciate the time and effort you have invested in reviewing our manuscript and for providing us with such valuable feedback. Your input has significantly improved the quality of our work. Should there be any additional concerns or further revisions, we will promptly address them.
Thank you for your consideration and valuable support.
Best regards,
Shuai Zhao
Yangzhou University
Reviewer 2 Report
Overall, this is a nicely done and well written publication. The study design is appropriate and apparently, the analyses were carefully performed. This manuscript shows rich and valuable content, which is within the journal’s scope.
However, before publication some points need to be clarified:
Line 4 – Please use superscript to identify the authors
Line 29 – please expand all acronyms used in abstract.
Line 51 – The gene names should be written in italics (Bcl-2).
Line 54 – the authors should ensure that they use the term “expression” in relation to genes only.
Line 59 – please provide any goal or hypothesis of this study.
Figure 2 – some elements of Figure 2 are too small to read, and therefore they are not informative enough.
Figure 4, 5 – please provide uncropped WB gels.
Line 289 – please provide method of geese euthanasia.
Line 289 – How many animals were used in this study? (what was n=?)
Line 289 – Please provide Ethic Committee agreement number.
Line 394 – please provide details of antibodies used (concentrations, country of origin etc.)
Author Response
Response to Reviewer 2 Comments
Dear Reviewer,
We sincerely appreciate your review of our manuscript and valuable feedback. We have carefully considered your comments and made the necessary revisions and additions. Please find our point-by-point response below:
Point 1: Line 4 - Please use superscript to identify the authors.
Response 1: We sincerely apologize for the oversight. We have now corrected this obvious mistake by using superscript to identify the authors (Line 4-5).
Point 2: Line 29 - Please expand all acronyms used in the abstract.
Response 2: Thank you for your valuable suggestion. We have expanded all acronyms in the abstract section and rectified similar instances throughout the manuscript (Line 19-22, 53, etc.).
Point 3: Line 51 - The gene names should be written in italics (e.g., Bcl-2).
Response 3: We appreciate your keen observation. The gene names have been appropriately written in italics throughout the manuscript (Line 23, 27, 186, 197, etc.).
Point 4: Line 54 - The authors should ensure that they use the term "expression" in relation to genes only.
Response Thank you for pointing this out. We have made the necessary corrections to ensure the term "expression" is used appropriately in relation to genes (Line 57).
Point 5: Line 59 - Please provide any goal or hypothesis of this study.
Response 5: The goal of our study is to employ LC-MS-MS-based proteomic analysis to investigate changes in protein phosphorylation in the ovaries of geese during the laying and brooding periods. Additionally, we aim to explore the molecular mechanisms of goose brooding behavior to address the issue of low goose production performance (Line 67-71).
Point 6: Figure 2 - Some elements are too small to read, and therefore, they are not informative enough.
Response 6: Our sincere apologies for the inconvenience caused. To solve this problem, we enlarged the text in the image and uploaded a high-resolution copy of the original image in the manuscript, ensuring that the content is clear when zoomed in and that the image remains editable (Line 137).
Point 7: Figure 4, 5 - Please provide uncropped WB gels.
Response 7: We have included the uncropped Western blot (WB) gels for Figures 4 and 5 in the supplementary materials (manuscript-supplementary.pdf).
Point 8: Line 289 - Please provide the method of geese euthanasia.
Response 8: Geese were euthanized using sodium pentobarbital (Line 316).
Point 9: Line 289 - How many animals were used in this study? (What was n=?)
Response 9: This study utilized three geese per experimental group (n=3) (Line 315).
Point 10: Line 289 - Please provide the Ethic Committee agreement number.
Response 10: We are truly grateful for your guidance on this matter, due to journal requirements, the ethical declaration is placed at the end of the manuscriptethical declaration is now included in the manuscript (Line 455-461).The Ethic Committee agreement number for this research is SYXK (Su) IACUC 2012-0029
Point 11: Line 394 - Please provide details of antibodies used (concentrations, country of origin, etc.).
Response 11: Details of the antibodies used in the experiments (manufacturer and antibody stock numbers) are listed in the manuscript (Lines 424-426).
We sincerely appreciate the time and effort you have invested in reviewing our manuscript and for providing us with such valuable feedback. Your input has significantly improved the quality of our work. Should there be any additional concerns or further revisions, we will promptly address them.
Thank you for your consideration and valuable support.
Best regards,
Shuai Zhao
Yangzhou University
Round 2
Reviewer 1 Report
Great job, congratulations!